# ADAPTIVE ACTIVATION-BASED STRUCTURED PRUNING

## ABSTRACT

Pruning is a promising approach to compress complex deep learning models in order to deploy them on resource-constrained edge devices. However, many existing pruning solutions are based on unstructured pruning, which yield models that cannot efficiently run on commodity hardware, and require users to manually explore and tune the pruning process, which is time consuming and often leads to sub-optimal results. To address these limitations, this paper presents an adaptive, activation-based, structured pruning approach to automatically and efficiently generate small, accurate, and hardware-efficient models that meet user requirements. First, it proposes iterative structured pruning using activation-based attention feature maps to effectively identify and prune unimportant filters. Then, it proposes adaptive pruning policies for automatically meeting the pruning objectives of accuracy-critical, memory-constrained, and latency-sensitive tasks. A comprehensive evaluation shows that the proposed method can substantially outperform the state-of-the-art structured pruning works on CIFAR-10 and ImageNet datasets. For example, on ResNet-56 with CIFAR-10, without any accuracy drop, our method achieves the largest parameter reduction (79.11%), outperforming the related works by 22.81% to 66.07%, and the largest FLOPs reduction (70.13%), outperforming the related works by 14.13% to 26.53%.

## 1 INTRODUCTION

Deep neural networks (DNNs) have substantial compute and memory requirements. As deep learning becomes pervasive and moves towards edge devices, DNN deployment becomes harder because of the mistmatch between resource-hungry DNNs and resource-constrained edge devices. DNN pruning is a promising approach (Li et al. (2016); Han et al. (2015); Molchanov et al. (2016); Theis et al. (2018); Renda et al. (2020)), which identifies the parameters (or weight elements) that do not contribute significantly to the accuracy and prunes them from the network. Recently, works based on the Lottery Ticket Hypothesis (LTH) have achieved great successes in creating smaller and more accurate models through iterative pruning with rewinding (Frankle & Carbin (2018)). However, LTH has only been shown to work successfully with unstructured pruning which, unfortunately leads to models with low sparsity and difficult to accelerate on commodity hardware such as CPUs and GPUs (e.g., Hill et al. (2017) shows directly applying NVDIA cuSPARSE on unstructured pruned models can lead to $60\times$ slowdown on GPU compared to dense kernels.) Moreover, most pruning methods require users to explore and adjust multiple hyper-parameters, e.g., with LTH-based iterative pruning, users need to determine how many parameters to prune in each round. Tuning the pruning process is time consuming and often leads to sub-optimal results.

We propose *activation-based*, *adaptive*, *iterative structured pruning* to find the "winning ticket" models that are at the same time hardware efficient and to automatically meet the users' model accuracy, size, and speed requirements. First, we propose an activation-based structured pruning method to identify and remove unimportant filters in an LTH-based iterative pruning (with rewinding) process. Specifically, we properly define an attention mapping function that takes a 2D activation feature maps of a filter as input, and outputs a 1D value used to indicate the importance of the filter. This approach is more effective than weight-value based filter pruning because activation-based attention values not only capture the features of inputs but also contain the information of convolution layers that act as feature detectors for prediction tasks. We then integrate this attention-

based method into the LTH-based iterative pruning framework to prune the filters in each round and find the winning ticket that is small, accurate, and hardware-efficient.

Second, we propose adaptive pruning that automatically optimizes the pruning process according to different user objectives. For latency-sensitive scenarios like interactive virtual assistants, we propose FLOPs-guaranteed pruning to achieve the best accuracy given the maximum amount of compute FLOPs; For memory-limited environments like embedded systems, we propose model-size-guaranteed pruning to achieve the best accuracy given the maximum amount of memory footprint; For accuracy-critical applications such as those on self-driving cars, we propose accuracy-guaranteed pruning to create the most resource-efficient model given the acceptable accuracy loss. Aiming for different targets, our method adaptively controls the pruning aggressiveness by adjusting the global threshold used to prune filters. Moreover, it considers the difference in each layer's contributions to the model's size and computational complexity and uses a per-layer threshold, calculated by dividing each layer's remaining parameters or FLOPs by the entire model's remaining parameters or FLOPs, to prune each layer with differentiated level of aggressiveness.

Our results outperform the related works significantly in all cases targeting accuracy loss, parameters reduction, and FLOPs reduction. For example, on ResNet-56 with CIFAR-10 dataset, without accuracy drop, our method achieves the largest parameter reduction (79.11%), outperforming the related works by 22.81% to 66.07%, and the largest FLOPs reduction (70.13%), outperforming the related works by 14.13% to 26.53%. In addition, our method enables a pruned model the reach 0.6% or 1.08% higher accuracy than the original model but with only 30% or 50%of the original's parameters. On ResNet-50 on ImageNet, for the same level of parameters and FLOPs reduction, our method achieves the smallest accuracy loss, lower than the related works by 0.08% to 3.21%; and for the same level of accuracy loss, our method reduces significantly more parameters (6.45% to 29.61% higher than related works) and more FLOPs (0.82% to 17.2% higher than related works).

## 2 BACKGROUND AND RELATED WORKS

**Unstructured vs. Structured Pruning.** Unstructured pruning (LeCun et al. (1990); Han et al. (2015); Molchanov et al. (2017)) is a fine-grained approach that prunes individual unimportant elements in weight tensors. It has less impact to model accuracy, compared to structured pruning, but unstructured pruned models are hard to accelerate on commodity hardware. Structured pruning is a coarse-grained approach that prunes entire regular regions of weight tensors according to some rule-based heuristics, such as L1-norm (Li et al. (2016)), average percentage of zero (Molchanov et al. (2016)), and other information considering the relationship between neighboring layers (Theis et al. (2018); Lee et al. (2018)). Compared to unstructured pruning, it is more difficult to prune a model without causing accuracy loss using structured pruning, because by removing entire regions, it might remove weight elements that are important to the final accuracy (Li et al. (2016)). However, structured pruned models can be mapped easily to general-purpose hardware and accelerated directly with off-the-shelf hardware and libraries (He et al. (2018b)).

**One Shot vs. Iterative Pruning.** One-shot pruning prunes a pre-trained model and then retrains it once, whereas iterative pruning prunes and retrains the model in multiple rounds. Both techniques can choose either structured or unstructured pruning techniques. Recently, works based on the Lottery Ticket Hypothesis (LTH) have achieved great successes in creating smaller and more accurate models through iterative pruning with rewinding (Frankle & Carbin (2018)). LTH posits that a dense randomly initialized network has a sub-network, termed as a *winning ticket*, which can achieve an accuracy comparable to the original network. At the beginning of each pruning round, it rewinds the weights and/or learning rate of the sub-network to some early epoch of the training phase of the original model to reduce the distance between the sub-network and original model and increase the change of finding the winning ticket. However, most of LTH-based works considered only unstructured pruning, e.g., Iterative Magnitude Pruning (IMP) (Frankle & Carbin (2018); Frankle et al. (2019)), which, as discussed above, is hardware-inefficient.

It is non-trivial to design an iterative pruning method with structured pruning. To understand the state of iterative structured pruning, we experimented with IMP's structured pruning counterpart—L1-norm structured pruning (ILP) (Li et al. (2016)) which removes entire filters depending on their L1-norm value. We observed that ILP cannot effectively prune a model while maintaining its accuracy, e.g., ILP can prune ResNet-50 by at most 11.5% of parameters when the maximum accuracy loss

---

**Algorithm 1** Adaptive Iterative Structured Pruning Algorithm

---

1: [Initialize] Initialize a network $f(x; M^0 \odot W_0^0)$ with initial mask $M^0 = \{0,1\}^{|W_0^0|}$ and Threshold $T[0]$
2: [Save weights] Train the network for $k$ epochs, yielding network $f(x; M^0 \odot W_k^0)$, and save weights $W_k^0$
3: [Train to converge] Train the network for $T - k$ epochs to converge, producing network $f(x; M^0 \odot W_T^0)$
4: **for** pruning round $r$ ($r \geqslant 1$) **do**
5:     [Prune] Prune filters from $W_T^{r-1}$ using $T[r]$, producing a mask $M^r$, and a network $f(x; M^r \odot W_T^{r-1})$
6:     [Rewind Weights] Reset the remaining filters to $W_k^0$ at epoch $k$, producing network $f(x; M^r \odot W_k^{r-1})$
7:     [Rewind Learning Rate] Reset the learning rate schedule to its state from epoch $k$
8:     [Retrain] Retrain the unpruned filters for $T - k$ epoch to converge, yielding network $f(x; M^r \odot W_T^r)$
9:     [Evaluate] Evaluate the retrained network $f(x; M^r \odot W_T^r)$ according to the target.
10:     [Reset Weights] If the target is not met, reset the weights to an earlier round
11:     [Adapt Threshold] Calculate Threshold $T[r+1]$ for the next pruning round
12: **end for**

---

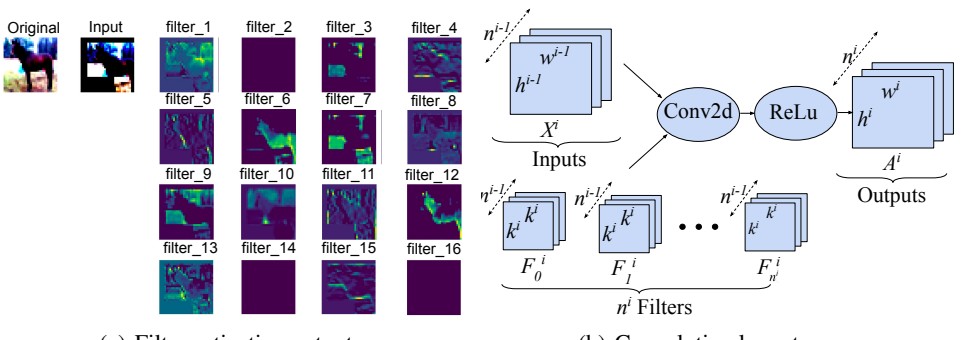

(a) Filter activation outputs          (b) Convolution layer tensors

Figure 1: Illustration of the (a) activation outputs of 16 filters of a conv2d layer, and (b) input and output tensors of a convolution layer. In (a), the left portion shows the original image and the image after data augmentation; and the right portion visualizes the activation outputs of each filter.

is limited to 1% on ImageNet. Therefore, directly applying iterative pruning with existing weight-magnitude based structured pruning methods does not produce accurate pruned models. In this paper, we study how to produce *small*, *accurate*, and *hardware-efficient* models based on *iterative structured pruning with rewinding*.

**Automatic Pruning.** For pruning to be useful in practice, it is important to automatically meet the pruning objectives for diverse ML applications and devices. Most pruning methods require users to explore and adjust multiple hyper-parameters, e.g., with LTH-based iterative pruning, users need to determine how many parameters to prune in each round. Tuning the pruning process is time consuming and often leads to sub-optimal results. Therefore, we study how to *automatically* adapt the pruning process in order to *meet the pruning objectives without user intervention*.

Some works (Zoph et al. (2018); Cai et al.; Ashok et al. (2017)) use reinforcement-learning algorithms to find computationally efficient architectures. These works are in the area of Neural Architecture Search (NAS), among which the most recent is AutoML for Model Compression (AMC) (He et al. (2018b)). AMC enables the model to arrive at the target speedup by limiting the action space (the sparsity ratio for each layer), and it finds out the limit of compression that offers no loss of accuracy by tweaking the reward function. But this method has to explore over a large search space of all available layer-wise sparsity, which is time consuming when neural networks are large and datasets are complicated.

## 3 METHODOLOGY

Algorithm 1 illustrates the overall flow of the proposed adaptive activation-based structured pruning. To represent pruning of weights, we use a mask $M^r \epsilon \{0,1\}^d$ for each weight tensor $W_t^r \epsilon R^d$, where $r$ is the pruning round number and $t$ is the training epoch. Therefore, the pruned network at the end

of training epoch $T$ is represented by the element-wise product $M^r \odot W_T^r$. The first three steps are to train the original model to completion, while saving the weights at Epoch $k$. Steps 4–10 represent a pruning round. Step 4 prunes the model (discussed in Section 3.1). Step 5 (optional) and 6 perform rewinding. Step 7 retrains the pruned model for the remaining $T - k$ epochs. Step 8 evaluates the pruned model according to the pruning target. If the target is not met, Step 9 resets the weights to an earlier round. Step 10 calculates the pruning threshold for the next pruning round following the adaptive pruning policy (discussed in Section 3.2).

## 3.1 ACTIVATION-BASED FILTER PRUNING

To realize iterative structured pruning, one can start with the state-of-the-art structured pruning methods and apply it iteratively. The widely used structured pruning method—L1-norm based structured pruning Renda et al. (2020) removes filters with the lowest L1-norm values, whereas the most recent method—Polarization-based structured pruning Zhuang et al. (2020) improves it by using a regularizer on scaling factors of filters and pruning filters whose scaling factors are below than a threshold. These two methods both assume that the *weight values* of a filter can be used as an indicator about the importance of that filter, much like how LTH uses weight values in unstructured pruning. However, we observe that weight-based structured pruning methods cannot produce accurate pruned models. For example, to prune ResNet-56 on CIFAR-10 with no loss in top-1 accuracy, L1-norm based structured pruning can achieve at most only $1.15\times$ model compression, and Polarization-norm based structured pruning can achieve at most only $1.89\times$ inference speedup. The reason is that, some filters, even though their weight values are small, can still produce useful non-zero activation values that are important for learning features during backpropagation. That is, filters with small values may have large activations.

We propose that the *activation values* of filters are more effective in finding unimportant filters to prune. Activations like ReLu enable non-linear operations, and enable convolutional layers to act as feature detectors. If an activation value is small, then its corresponding feature detector is not important for prediction tasks. So activation values, i.e., the intermediate output tensors after the non-linear activation, not only detect features of training dataset, but also contain the information of convolution layers that act as feature detectors for prediction tasks. We present a visual motivation in Figure 1(a). The figure shows the activation output of 16 filters of a convolution layer on one input image. The first image on the left is the original image, and the second image is the input features after data augmentation. We observe that some filters extract image features with high activation patterns, e.g., the 6th and 12th filters. In comparison, the activation outputs of some filters are close to zero, such as the 2nd, 14th, and 16th. Therefore, from visual inspection, removing filters with weak activation patterns is likely to have low impact on the final accuracy of the pruned model.

There is a natural connection between our activation-based pruning approach and the related attention-based knowledge transfer works (Zagoruyko & Komodakis (2016)). Attention is a mapping function used to calculate the statistics of each element in the activations over the channel dimension. By minimizing the difference of the activation-based attention maps from intermediate layers between a teacher model and a student model, attention transfer enables the student to imitate the behaviour of the teacher. Our proposed pruning method builds upon the same theory that activation-based attention is a good indicator of filters regarding their ability to capture features, and it addresses the new challenges in using activations to guide automatic structured pruning.

In the following, we describe how to prune filters based on its activation feature maps in each round of the iterative structured pruning process. Figure 1(b) shows the inputs and outputs of a 2D convolution layer (referred to as conv2d), followed by an activation layer. For the $i$th conv2d layer, let $X^i \epsilon R^{n^{i-1} \times h^{i-1} \times w^{i-1}}$ denote the input features, and $F_j^i \epsilon R^{n^{i-1} \times k^i \times k^i}$ be the $j$th filter, where $h^{i-1}$ and $w^{i-1}$ are the height and width of the input features, respectively, $n^{i-1}$ is the number of input channels, $n^i$ is the number of output channels, and $k^i$ is the kernel size of the filter. The activation of the $j$th filter $F_j^i$ after ReLu mapping is therefore denoted by $A_j^i \epsilon R^{h^i \times w^i}$.

The attention mapping function takes a 2D activation $A_j^i \epsilon R^{h^i \times w^i}$ of filter $F_j^i$ as input, and outputs a 1D value which will be used as an indicator of the importance of filters. We consider three forms of activation-based attention maps, where $p \geqslant 1$ and $a_{k,l}^i$ denotes every element of $A_j^i$: 1) Mean of attention values raised to the power of $p$: $F_{mean}(A_j^i) = \frac{1}{h^i \times w^i} \sum_{k=1}^{h^i} \sum_{l=1}^{w^i} \left| a_{k,l}^i \right|^p$; 2) Max of atten-

---

**Algorithm 2** Accuracy-guaranteed Adaptive Pruning

---

1: **Input:** Target Accuracy Loss $AccLossTarget$
2: **Output:** A small pruned model with an acceptable accuracy
3: Initialize: $T = 0.0$, $\lambda = 0.005$.
4: **for** pruning round $r$ ($r \geqslant 1$) **do**
5:     Prune the model using $T[r]$ (Refer to Algorithm 3)
6:     Train the pruned model, evaluate its accuracy $Acc[r]$
7:     Calculate the accuracy loss $AccLoss[r]$: $AccLoss[r] = Acc[0] - Acc[r]$
8:     **if** $AccLoss[r] < AccLossTarget$ **then**
9:         **if** the changes of model size are within 0.1% for several rounds **then**
10:             Terminate
11:         **else**
12:             $\lambda[r+1] = \lambda[r]$
13:             $T[r+1] = T[r] + \lambda[r+1]$
14:         **end if**
15:     **else**
16:         Find the last acceptable round $k$
17:         **if** $k$ has been used to roll back for several times **then**
18:             Mark $k$ as unacceptable
19:             Go to Step 15
20:         **else**
21:             Roll back model weights to round $k$
22:             $\lambda[r+1] = \lambda[r]/2.0^{(N+1)}$ ($N$ is the number of times for rolling back to round $k$)
23:             $T[r+1] = T[k] + \lambda[r+1]$
24:         **end if**
25:     **end if**
26: **end for**

---

tion values raised to the power of $p$: $F_{max}(A_j^i) = max_{l=1,h^i \times w^i} \left| a_{k,l}^i \right|^p$; and 3) Sum of attention values raised to the power of $p$: $F_{sum}(A_j^i) = \sum_{k=1}^{h^i} \sum_{l=1}^{w^i} \left| a_{k,l}^i \right|^p$. We choose $F_{mean}(A_j^i)$ with $p$ equals to 1 as the indicator to identify and prune unimportant filters, and our method removes the filters whose attention value is lower than the pruning threshold. See Section 5 for an ablation study on these choices.

## 3.2 ADAPTIVE ITERATIVE PRUNING

Our approach to pruning is to automatically and efficiently generate a pruned model that meets the users' different objectives. Automatic pruning means that users do not have to figure out how to configure the pruning process. Efficient pruning means that the pruning process should produce the user-desired model as quickly as possible. Users' pruning objectives can vary depending on the usage scenarios: 1) Accuracy-critical tasks, like those used by self-driving cars have stringent accuracy requirement, which is critical for safety, but do not have strict limits on their computing and storage usages; 2) Memory-constrained tasks, like those deployed on microcontrollers have very limited available memory to store the models but do not have strict accuracy requirement; and 3) Latency-sensitive tasks, like those employed by virtual assistants where timely responses are desirable but accuracy is not a hard constraint.

In order to achieve automatic and efficient structured pruning, we propose three adaptive pruning policies to provide 1) Accuracy-guaranteed Adaptive Pruning which produces the most resource-efficient model with the acceptable accuracy loss; 2) Memory-constrained Adaptive Pruning which generates the most accurate model within a given memory footprint; and 3) FLOPs-constrained Adaptive Pruning which creates the most accurate model within a given computational intensity. Specifically, our proposed adaptive pruning method automatically adjusts the global threshold ($T$), used in our iterative structured pruning algorithm (Algorithm 1) to quickly find the model that meets the pruning objective. Other objectives (e.g., limiting a model's energy consumption) can also be supported by simply plugging in the relevant metrics into our adaptive pruning algorithm. We take Accuracy-guaranteed Adaptive Pruning, described in Algorithm 2, as an example to show the procedure of adaptive pruning. Memory-constrained and FLOPs-constrained Adaptive Pruning algorithms are included in Appendix A.3 and A.4, respectively.

---

**Algorithm 3** Layer-aware Threshold Adjustment

---

1: **for** the layer $i$ in a pruning round $r$ **do**
2:    Calculate the number of parameters of unpruned filters of the current layer and the whole model, respectively: $N^i[r] = n^{(i-1)}[r] \times k^i[r] \times k^i[r] \times n^i[r]$, $N^{Total}[r] = \sum_{i=1}^{i=M} n^{(i-1)}[r] \times k^i[r] \times k^i[r] \times n^i[r]$, where $M$ is the number of convolution layers, and for other notations refer to Section 3.1.
3:    Calculate the threshold $T^i[r]$ of the current layer: $T^i[r] = T[r] \times \frac{N^i[r]}{N^{Total}[r]}$.
4:    For each filter $F_j^i$ in $F^i$, calculate its attention: $F_{mean}(A_j^i) = \frac{1}{h^i \times w^i} \sum_{k=1}^{h^i} \sum_{l=1}^{w^i} |a_{k,l}^i|^p$.
5:    Prune the filter if its attention is not larger than threshold $T^i[r]$, i.e., set $M_j^i = 0$ if $F_{mean}(A_j^i) \leqslant T^i[r]$; Otherwise, set $M_j^i = 1$
6: **end for**

---

In Algorithm 2, the objective is to guarantee the accuracy loss while minimizing the model size (the algorithm for minimizing the model FLOPs is similar). In the algorithm, $T$ controls the aggressiveness of pruning, and $\lambda$ determines the increment of $T$ at each pruning round. Pruning starts conservatively, with $T$ initialized to $0$, so that only completely useless filters that cannot capture any features are pruned. After each round of pruning, if the model accuracy loss is below than the target accuracy loss, it is considered "acceptable", and the algorithm increases the aggressiveness of pruning by incrementing $T$ by $\lambda$. As pruning becomes increasingly aggressive, the accuracy eventually drops below the target accuracy at a certain round which is considered "unacceptable". When this happens, our algorithm rolls back the model weights and pruning threshold to the last acceptable round where the accuracy loss is within the target, and restarts the pruning from there but more conservatively—it increases the threshold more slowly by cutting the $\lambda$ value by half. If this still does not lead to an acceptable round, the algorithm cuts $\lambda$ by half again and restarts again. If after several trials, the accuracy loss is still not acceptable, the algorithm rolls back even further and restarts from an earlier round. The rationale behind this adaptive algorithm is that the aggressiveness of pruning should accelerate when the model is far from the pruning target, and decelerate when it is close to the target. Eventually, the changes of model size converges (when the changes of the number of parameters is within $0.1\%$ for several rounds) and the algorithm terminates.

## 3.3 LAYER-AWARE THRESHOLD ADJUSTMENT

While adapting the global pruning threshold using the above discussed policies, our pruning method further considers the difference in each layer's contribution to model size and complexity and use differentiated layer-specific thresholds to prune the layers. As illustrated in Figure 1(b), in terms of the contribution to model size, the number of parameters of layer $i$ can be estimated as $N^i = n^{(i-1)} \times k^i \times k^i \times n^i$; in terms of the contribution to computational complexity, the number of FLOPs of layer $i$ can be estimated as $2 \times h^i \times w^i \times N^i$. A layer that contributes more to the model's size or FLOPs is more likely to have redundant filters that can be pruned without affecting the model's accuracy. Therefore, to effectively prune a model while maintaining its accuracy, we need to treat each layer differently at each round of the iterative pruning process based on its current contributions to the model size and complexity. Specifically, our adaptive pruning method calculates a weight for each layer based on its contribution and then uses this weight to adjust the current global threshold and derive a local threshold for the layer. If the goal is to reduce model size, the weight is calculated as each layer's number of parameters divided by the model's total number of parameters; if the goal is to reduce model computational complexity, the weight is calculated as each layer's FLOPs divided by the model's total FLOPs. These layer-specific thresholds are then used to prune the layers in the current pruning round.

Assuming the goal is to reduce model size, the procedure in a pruning round $r$ for the $i$th conv2d layer is shown in Algorithm 3 (the algorithm for reducing model FLOPs is similar). For pruning, we introduce a mask $M_j^i \epsilon \{0,1\}^{n^{i-1} \times k^i \times k^i}$ for each filter $F_j^i$. The effective weights of the convolution layer after pruning is the element-wise product of the filter and mask—$F_j^i \odot M_j^i$. For a filter, all values of this mask is either $0$ or $1$, thus either keeping or pruning the entire filter.

Table 1: Comparing the proposed method with state-of-the-art structured pruning methods on the CIFAR-10 dataset. The baseline Top-1 accuracy of ResNet-56 and ResNet-50 on CIFAR-10 is 92.84%, 91.83% (5 runs), respectively. The numbers of the related works are cited from their papers.

| Model | Target | Target Level | Method | Acc. ↓ (%) | Params. ↓ (%) | FLOPs. ↓ (%) |
|---|---|---|---|---|---|---|
| ResNet-56 | Acc. ↓ (%) | 0% | SCOP | 0.06 | 56.30 | 56.00 |
| | | | HRank | 0.09 | 42.40 | 50.00 |
| | | | PFF | 0.03 | 42.60 | 43.60 |
| | | | ILP | 0.00 | 13.04 | - |
| | | | PPR | -0.03 | - | 47.00 |
| | | | EagleEye | -1.4 | - | 50.41 |
| | | | **Ours-1** | **-0.33** | **79.11** | **-** |
| | | | **Ours-2** | **-0.08** | **-** | **70.13** |
| | | 1% | RFR | 1.00 | 50.00 | - |
| | | | ILP | 1.00 | 41.18 | - |
| | | | GAL | 0.52 | 44.80 | 48.50 |
| | | | **Ours-1** | **0.86** | **88.23** | **-** |
| | | | **Ours-2** | **0.77** | **-** | **81.19** |
| | Params. ↓ (%) | 70% | DCP | -0.01 | 70.30 | 47.10 |
| | | | GBN | 0.03 | 66.70 | 70.30 |
| | | | HRank | 2.38 | 68.10 | 74.10 |
| | | | GDP | -0.35 | 65.64 | - |
| | | | **Ours** | **-0.6** | **71.57** | **-** |
| | | 50% | CP | 1.00 | - | 50.00 |
| | | | SFP | 1.33 | 50.60 | 52.60 |
| | | | FPGM | 0.10 | 50.60 | 52.60 |
| | | | GDP | -0.07 | 53.35 | - |
| | | | **Ours** | **-1.08** | **53.3** | **-** |
| | FLOPs. ↓ (%) | 75% | HRank | 2.38 | 68.10 | 74.10 |
| | | | DH | 2.54 | - | 71.00 |
| | | | PPR | 1.17 | - | 71.00 |
| | | | **Ours** | **0.3** | **-** | **71.44** |
| | | 55% | CP | 1 | - | 50.00 |
| | | | SFP | 1.33 | 50.60 | 52.60 |
| | | | FPGM | 0.10 | 50.60 | 52.60 |
| | | | AMC | 0.90 | - | 50.00 |
| | | | **Ours** | **-0.63** | **-** | **52.92** |
| ResNet-50 | Params. ↓ (%) | 60% | AMC | -0.02 | 60.00 | - |
| | | | **Ours** | **-0.86** | **64.81** | **-** |
| VGG-16 | Acc. ↓ (%) | 0% | PFS | -0.19 | 50 | - |
| | | | SCP | 0.06 | 66.23 | - |
| | | | VCNNP | 0.07 | 60.9 | - |
| | | | Hinge | 0.43 | 39.07 | - |
| | | | HRank | 0.53 | 53.6 | - |
| | | | **Ours** | **-0.16** | **72.85** | **61.17** |
| | Params. ↓ (%) | 70% | GDP | -0.1 | 69.45 | - |
| | | | **Ours** | **-0.29** | **70.54** | **41.72** |
| VGG-19 | Acc. ↓ (%) | 0% | Eigendamage | 0.19 | 78.18 | 37.13 |
| | | | NN Slimming | 1.33 | 80.07 | 42.65 |
| | | | **Ours** | **-0.26** | **85.99** | **56.22** |
| | FLOPs. ↓ (%) | 85% | Eigendamage | 1.88 | - | 86.51 |
| | | | **Ours** | **1.81** | **-** | **89.02** |
| MobileNetV2 | Acc. ↓ (%) | 0% | GDP | -0.26 | 46.22 | - |
| | | | MDP | -0.12 | 28.71 | - |
| | | | **Ours** | **-0.54** | **60.16** | **35.93** |
| | Params. ↓ (%) | 40% | SCOP | 0.24 | 40.30 | - |
| | | | DMC | -0.26 | 40 | - |
| | | | **Ours** | **-0.8** | **43.77** | **78.65** |

## 4 EVALUATION

### 4.1 CIFAR-10 RESULTS

Table 1 lists the results from CIFAR-10 on ResNet-56, ResNet-50, VGG-16, VGG-19, and MobileNetV2. We compare our method with the state-of-the-art works, including **SCOP** (Tang et al. (2020)), **HRank** (Lin et al. (2020)), **ILP** (Renda et al. (2020)), **PPR** (Zhuang et al. (2020)), **RFR** (He et al. (2017)), **GAL** (Lin et al. (2019)), **DCP** (Zhuang et al. (2018)), **GBN** (You et al. (2019)), **CP** (He et al. (2017)), **SFP** (He et al. (2018a)), **FPGM** (He et al. (2019)), DeepHoyer (**DH**) (Yang et al. (2019)), **AMC** (He et al. (2018b)), **PFS** (Wang et al. (2020)), **SCP** (Kang & Han (2020)), **VCNNP** (Zhao et al. (2019)), **Hinge** (Li et al. (2020)), **GDP** (Guo et al. (2021)), **Eigendamage** (Wang et al. (2019)), **NN Slimming** (Liu et al. (2017)), **MDP** (Guo et al. (2020)), and **DMC** (Gao et al. (2020)). Note that a negative accuracy drop means that the pruned model achieves better accu-

Table 2: Comparing the proposed method with state-of-the-art structured pruning methods on ResNet-50 with ImageNet dataset, and VGG-19 on Tiny-ImageNet. The result of ILP is from our implementation, and all the other numbers of the related works are cited from their papers.

| Model | Targets | Targets Level | Method | Acc. ↓ (%) | Params. ↓ (%) | FLOPs ↓ (%) |
|---|---|---|---|---|---|---|
| ResNet-50 | Acc. ↓ (%) | 0% | PFP-A | 0.22 | 18.10 | 10.80 |
| | | | **Ours-1** | **-0.12** | **24.55** | **-** |
| | | | **Ours-2** | **0.18** | **-** | **11.62** |
| | | 1% | SSS-41 | 0.68 | 0.78 | 15.06 |
| | | | ILP | 1.00 | 11.5 | - |
| | | | **Ours-1** | **0.64** | **30.39** | **-** |
| | | | **Ours-2** | **0.83** | **-** | **32.26** |
| | Params. ↓ (%) | 40% | Hrank | 1.17 | 36.70 | 43.70 |
| | | | SSS-26 | 4.30 | 38.82 | 43.04 |
| | | | **Ours** | **1.09** | **37.17** | **-** |
| | | 30% | PFP-B | 0.92 | 30.10 | 44.00 |
| | | | **Ours** | **0.48** | **29.42** | **-** |
| | FLOPs. ↓ (%) | 30% | SSS-32 | 1.94 | 27.06 | 31.08 |
| | | | **Ours** | **0.57** | **-** | **31.88** |
| VGG-19 | Acc. ↓ (%) | 3% | EigenDamage | 3.36 | 61.87 | 66.21 |
| | | | **Ours** | **2.61** | **72.58** | **78.97** |
| | Params. ↓ (%) | 60% | NN Slimmin | 10.66 | 60.14 | - |
| | | | **Ours** | **-0.34** | **60.21** | **-** |

racy than its unpruned baseline model. "Ours-1" and "Ours-2" denote the cases where the pruning objective is set to minimize model size and FLOPs, respectively, while meeting the accuracy target.

In all cases targeting accuracy, model size, and compute intensity, our method significantly outperforms the recent related works. For example, on ResNet-56, without accuracy drop, our method achieves the largest parameter reduction (79.11%), outperforming the related works by 22.81% to 66.07%, and the largest FLOPs reduction (70.13%), outperforming the related works by 14.13% to 26.53%. With 70% of parameter reduction, our method achieves the smallest accuracy loss (-0.6%), outperforming the related works by 0.59% to 2.98%. Also, with 75% of FLOPs reduction, our method achieves the smallest accuracy loss (0.3%), outperforming the related works by 0.87% to 2.24%. Note that the proposed method also produces a pruned model that reaches 0.6% or 1.08% higher accuracy than the original model but with only 30% or 50%, respectively, of the original's parameters. Such small and accurate models are useful for many real-world applications.

On VGG-16, without accuracy drop, our method achieves the largest parameter reduction (72.85%), outperforming the related works by 6.62% to 33.78%; With 70% of parameter reduction, the accuracy loss achieved by our method is lower than GDP by 0.19%. On VGG-19, with no accuracy loss, our method achieves the largest parameter reduction (85.99%) which outperforms the related works by 5.92% to 7.81%, and the largest FLOPs reduction (56.22%) which outperforms the related works by 13.57% to 19.09%; With 85% of FLOPs reduction, the accuracy loss achieved by our method is lower than EigenDamage by 0.07%.

On MobileNet, without accuracy drop, our method achieves the largest parameter reduction (60.16%), outperforming GDP and MDP by 13.94% and 31.45%, respectively; With 40% of parameter reduction, the accuracy loss achieved by our method is lower than SCOP and DMC by 1.04% and 0.54%, respectively.

The results validate that activation-based pruning is more effective than weight magnitude based pruning in finding unimportant filters, as discussed in Section 3.1. For example, on ResNet-56, with 1% of accuracy loss, the amount of parameter reduction achieved by our method is higher than that of ILP (Renda et al. (2020)) by 34.51%; and with 75% FLOPs reduction, the accuracy achieved by our method is higher than PPR (Zhuang et al. (2020)) by 0.87%.

The results confirm that our method can achieve better results that the state-of-the-art automatic pruning methods, AMC (He et al. (2018b)). For example, when targeting at about 55% of FLOPs reduction on ResNet-56, our method achieves 1.53% higher accuracy than AMC; and when targeting at about 60% of parameters reduction on ResNet-50, our method achieves 0.84% higher accuracy than AMC. Our activation-based adaptive pruning can directly target unimportant filters in the model, whereas AMC has to use reinforcement learning to explore the whole network.

## 4.2 IMAGENET RESULTS

Table 2 shows the results of of ResNet-50 on ImageNet dataset and VGG-19 on Tiny-ImageNet dataset, comparing our method with **PFB** (Liebenwein et al. (2019)), **HRank** (Lin et al. (2020)),

Table 3: Results of one-shot pruning with different types of attention mapping functions and L1 Norm for VGG-16 on CIFAR-10 dataset.

| Method | Acc. ↓ (%) |
|---|---|
| **Attention Mean (p=1)** | **0.38** |
| Attention Mean (p=2) | 0.42 |
| Attention Mean (p=4) | 0.47 |
| Attention Sum (p=1) | 0.63 |
| Attention Max (p=1) | 0.51 |
| L1 Norm | 0.48 |

Table 4: Results of the adaptive pruning strategy and non-adaptive iterative pruning strategies for VGG-16 on CIFAR-10 dataset.

| Method | Acc. ↓ (%) | Params. ↓ (%) |
|---|---|---|
| IAP | 0.9 | 69.83 |
| ILP | 1.1 | 69.83 |
| **Ours** | **-0.29** | **70.54** |

**ILP** (Renda et al. (2020)), Sparse Structure Selection (**SSS**) (Huang & Wang (2018)), **NN Slimming** (Liu et al. (2017)), and **Eigendamage** (Wang et al. (2019)). "Ours-1" and "Ours-2" denote the cases where the pruning objective is set to minimize the number of parameters and FLOPs, respectively, while meeting the accuracy target. On ImageNet, for the same level of accuracy loss, our method reduces significantly more parameters (6.45% to 29.61% higher than the related works) and more FLOPs (0.82% to 17.2% higher than the related works). For the same level of parameters or FLOPs reduction, our method achieves the smallest accuracy loss, lower than the related works by 0.08% to 3.21%. On Tiny-ImageNet, for the same level of parameters reduction, our method achieves significantly lower accuracy loss than NN Slimming by 11%; For the same level of accuracy loss, our method achieves higher parameter reduction (72.58% vs 61.87%), and higher FLOPs reduction (78.97% vs 66.21%) than EigenDamage.

## 5 ABLATION STUDY

**The effect of attention mapping functions.** Table 3 shows the Top-1 accuracy loss of the VGG-16 pruned by one-shot pruning with different types of attention mapping functions on CIFAR-10. The parameter reduction is 57.73%, and the accuracy of the original model is 93.63% (5 trails). First, we analyze three forms of activation-based attention maps with $p$ equal to 1 (discussed in Section 3.1), noted as Attention Mean ($p = 1$), Attention Sum ($p = 1$), Attention Max ($p = 1$). Attention Mean leads to the lowest accuracy loss, lower than Attention Sum and Attention Max by 0.25% and 0.13%, respectively. Second, we analyze Attention Mean with difference values of $p$ ($p = 1, 2, 4$). When $p$ is set to 1, it leads to the lowest accuracy loss (0.38% vs 0.42% and 0.47%). Also, all the forms of Attention Mean outperforms L1-Norm, which validates that the proposed activation-based attention mapping function is more efficient than weight magnitude based methods to evaluate the importance of filters.

**The effect of adaptive pruning.** We compare the proposed adaptive pruning strategy with non-adaptive iterative pruning strategies that prune a fixed percentage of filters in each pruning round. One baseline prunes a fixed percentage of filters with lowest weight values, noted as Iterative L1-Norm Pruning (ILP); the other prunes a fxied percentage of filters using proposed attention mapping function with $p$ equal to 1, noted as Iterative Attention Pruning (IAP). Table 4 compares the Top-1 accuracy loss of the VGG-16 model with sparsity of 70% pruned by IAP, ILP, and the proposed adaptive pruning strategy on CIFAR-10. For IAP and ILP, the pruning rate is set to 5%, that is, 5% of filters are pruned in each pruning round. With 70% reduced parameters, the proposed adaptive pruning strategy leads to a higher accuracy than IAP and ILP by 1.19% and 1.39%, respectively.

## 6 CONCLUSIONS

This paper proposes an adaptive, activation-based, iterative structured pruning approach to automatically and efficiently generate small, accurate, and hardware-efficient models that can meet diverse user requirements. We show that activation-based attention value is a more precise indicator to identify unimportant filters to prune than the commonly used weight magnitude value. We also offer an effective way to perform structured pruning in an iterative pruning process, leveraging the LTH theory to find small and accurate sub-networks that are at the same time hardware efficient. Finally, we argue that automatic pruning is essential for this method to be useful in practice, and proposes an adaptive pruning method that can automatically meet user-specified objectives in terms of model accuracy, size, and inference speed but without user intervention. Our results confirm that the proposed method outperforms existing structured pruning approaches with a large margin.

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

# A    APPENDIX

## A.1    IMPLEMENTATION DETAILS

We implemented our proposed method on PyTorch version 1.6.0. For ResNet models on CIFAR-10, the learning rate is set to 0.1 initially and decays with a rate of 0.1 at the epoch 91 and 136. Weight decay is set to 0.0002. For ResNet-50 on ImageNet, the learning rate increases to 0.256 in a warmup mechanism during the first 5 epochs, and decays with a factor of 0.1 at epochs 30, 60, 80 (Renda et al. (2020); Frankle et al. (2019)). Nesterov SGD optimizer is used with a momentum of 0.9 for all models. The simple data augmentation (random crop and random horizontal flip) is used for all training images. Learning rate rewinding is used for iterative pruning, which rewinds to the epoch at roughly 60% of the total training duration (see Appendix A.5 for an analysis of the effect of rewinding epochs on the accuracy of the pruned models). For adaptive pruning, $T$ and $\lambda$ are initialized to 0.0 and 0.005, respectively.

## A.2    THE EFFECT OF ATTENTIONS

Figure 2 shows the attention of each filter of the first convolution layer of ResNet-50 on ImageNet with different values of $p$ (p=1, 2, 4). The setting where $p$ is equal to 1 tends to be best, since iit promotes the effectiveness of the pruning by enabling the gap between the mean values of the useful filters and useless filters to be large.

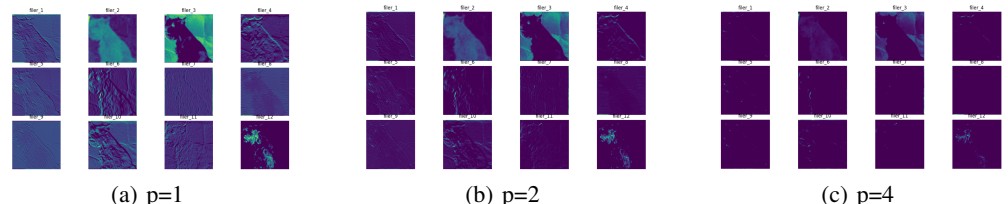

(a) p=1                    (b) p=2                    (c) p=4

Figure 2: Attentions of each filter of the first convolution layer of ResNet-50 on ImageNet with different values of $p$ ($p = 1, 2, 4$).

## A.3    MEMORY-CONSTRAINED ADAPTIVE PRUNING

The Memory-constrained Adaptive Pruning Algorithm is shown in Algorithm 4.

## A.4    FLOPS-CONSTRAINED ADAPTIVE PRUNING

The FLOPs-constrained Adaptive Pruning Algorithm is shown in Algorithm 5.

## A.5    THE EFFECT OF REWINDING EPOCH

In order to understand how the rewinding impacts accuracy of the pruned models, we analyze *stability to pruning* as proposed by Frankle et al. (2019), which is defined as the L2 distance between the masked weights of the pruned network and the original network at the end of training. We validate the observations proposed by Frankle et al. (2019) that for deep networks, rewinding to very early stages is sub-optimal as the network has not learned considerably by then; and rewinding to very late training stages is also sub-optimal because there is not enough time to retrain. Specifically, Figure 3(a) shows the Top-1 test accuracy of the pruned ResNet-50 with 83.74% remaining parameters when learning rate is rewound to different epochs, and Figure 3(b) shows the stability values at the corresponding rewinding epochs. We observe that there is a region, 65 to 80 epochs, where

---

**Algorithm 4** Memory-constrained Adaptive Pruning

---

1: **Input:** Target Parameters Reduction $ParamTarget$
2: **Output:** A small pruned model with an acceptable model size
3: Initialize: $T = 0.0, \lambda = 0.005$.
4: **for** pruning round $r$ $(r \geqslant 1)$ **do**
5:     Prune the model using $T[r]$ (Refer to Algorithm 3)
6:     Train the pruned model, calculate its remaining number of parameters $Param[r]$
7:     Calculate the parameters reduction: $ParamRed[r]$: $ParamRed[r] = Param[0] - Param[r]$
8:     **if** $ParamRed[r] < ParamTarget$ **then**
9:         **if** the changes of model size are within 0.1% for several rounds **then**
10:             Terminate
11:         **else**
12:             $\lambda[r+1] = \lambda[r]$
13:             $T[r+1] = T[r] + \lambda[r+1]$
14:         **end if**
15:     **else**
16:         Find the last acceptable round $k$
17:         **if** $k$ has been used to roll back for several times **then**
18:             Mark $k$ as unacceptable
19:             Go to Step 15
20:         **else**
21:             Roll back model weights to round $k$
22:             $\lambda[r+1] = \lambda[r]/2.0^{(N+1)}$ ($N$ is the number of times for rolling back to round $k$)
23:             $T[r+1] = T[k] + \lambda[r+1]$
24:         **end if**
25:     **end if**
26: **end for**

---

**Algorithm 5** FLOPs-constrained Adaptive Pruning

---

1: **Input:** Target FLOPs Reduction $FLOPsTarget$
2: **Output:** A small pruned model with an acceptable FLOPs
3: Initialize: $T = 0.0, \lambda = 0.005$.
4: **for** pruning round $r$ $(r \geqslant 1)$ **do**
5:     Prune the model using $T[r]$ (Refer to Algorithm 3)
6:     Train the pruned model, calculate its remaining FLOPs $FLOPs[r]$
7:     Calculate the parameters reduction: $FLOPsRed[r]$: $FLOPsRed[r] = FLOPs[0] - FLOPs[r]$
8:     **if** $FLOPsRed[r] < FLOPsTarget$ **then**
9:         **if** the changes of FLOPs are within 0.1% for several rounds **then**
10:             Terminate
11:         **else**
12:             $\lambda[r+1] = \lambda[r]$
13:             $T[r+1] = T[r] + \lambda[r+1]$
14:         **end if**
15:     **else**
16:         Find the last acceptable round $k$
17:         **if** $k$ has been used to roll back for several times **then**
18:             Mark $k$ as unacceptable
19:             Go to Step 15
20:         **else**
21:             Roll back model weights to round $k$
22:             $\lambda[r+1] = \lambda[r]/2.0^{(N+1)}$ ($N$ is the number of times for rolling back to round $k$)
23:             $T[r+1] = T[k] + \lambda[r+1]$
24:         **end if**
25:     **end if**
26: **end for**

---

the resulting accuracy is high. We find that L2 distance closely follows this pattern, showing large distance for early training epochs and small distance for later training epochs. Our findings show that rewinding to 75%-90% of training time leads to good accuracy.

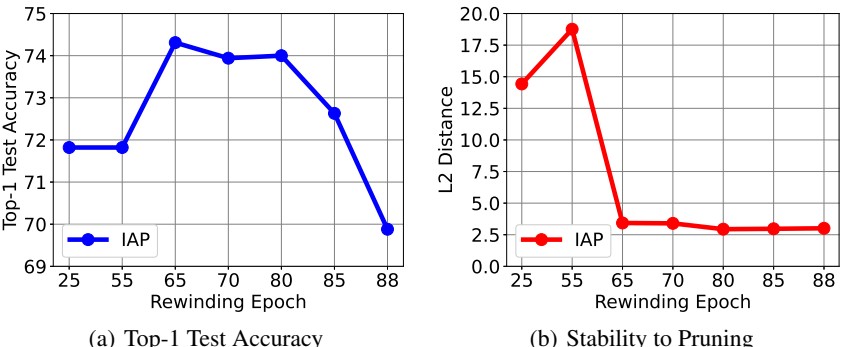

(a) Top-1 Test Accuracy          (b) Stability to Pruning

Figure 3: The effect of the rewinding epoch (x-axis) on (a) Top-1 test accuracy, and (b) pruning stability, for pruned ResNet-50, when 83.74% of parameters are remaining.

## A.6 THE EFFECT OF ITERATIVE PRUNING

To illustrate the effectiveness of our proposed adaptive pruning, in Figure 4, we show how the accuracy loss and parameter reduction change over the pruning rounds as the pruning threshold adapts according to Algorithm 2, in the experiment of pruning ResNet-56 on CIFAR-10, given the target of 1% accuracy loss. From Round 1 to Round 24, the accuracy loss of the pruned model is lower than the target accuracy loss (1%), so the algorithm increases the pruning aggressiveness gradually by increasing the threshold. At Round 25, the accuracy loss exceeds the target, so the algorithm rolls back the model weights and the pruning threshold to Round 24, and restart the pruning from there more conservatively. The above process repeats until after Round 39, the model size converges, and the algorithm terminates at reducing totally 88.23% of parameters.

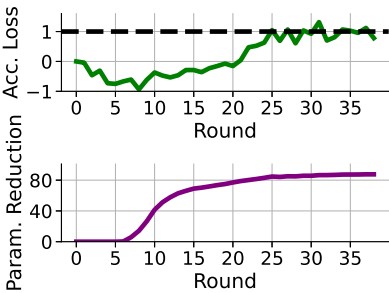

Figure 4: Illustration of the changes of the accuracy loss and parameter reduction over the pruning rounds as the pruning threshold adapts when pruning ResNet-56 with CIFAR-10, given the target of 1% Top-1 accuracy loss.

