# OpenReview forum: "Adaptive Activation-based Structured Pruning"
_ICLR.cc/2022/Conference — ICLR 2022 Submitted_

### Official Review · Reviewer_XNxE · 2021-11-01

**Correctness:** 3
**Technical Novelty And Significance:** 2
**Empirical Novelty And Significance:** 2
**Recommendation:** 3
**Confidence:** 5

**Main Review:**

Strength:
+ The proposed method is simple and easy to follow.
+ This paper proposes three modes for structured pruning, including accuracy-guaranteed adaptive pruning, memory-constrained adaptive pruning, and FLOPs-constrained adaptive pruning.

Weakness:
- The contribution of the evaluation for the importance of filter, since activation-based attention maps are similar to attention score in [Sergey Zagoruyko, ICLR 16] and the activation-based score is also proposed in many related works, such as APoZ [1] and HRank. Please clarify the difference between these works and the proposed methods.
- The rationale of calculating the threshold T in each layer. T in layer-aware threshold adjustment is directly related to the percentage of each layer on parameters or FLOPs. Moreover, the typo in line 3 of algorithm N^i[r]->T^i[r].
- Lack of some important experimental evaluation: 1. The models selected to compress are all based resnet architectures, how about other architectures, especially light backbones (e.g. mobilenets); 2. The effect of attention score and pruning strategy should be discussed in ablation study; 3. The actual speedup of the pruned models should be evaluated if minimizing FLOPs.

[1] Network trimming: A data-driven neuron pruning approach towards efficient deep architectures, arXiv preprint arXiv:1607.03250, 2016.


**Summary Of The Paper:**

This paper proposes iterative structured pruning methods using activation-based attention feature maps and an adaptive threshold selection strategy. Inspired by attention transfer, Activation-based attention feature maps are constructed as the important evaluation of filters in each layer. Adaptive threshold selection strategy decides the number of removed filters, which satisfies one of three adaptive pruning policies. Experimental results on CIFAR-10 and ImageNet with ResNet architectures show performance gains over several state-of-the-art methods.

**Summary Of The Review:**

Although the proposed method is simple yet effective on ResNets, the contribution is limited and experimental evaluation is not comprehensive. I tend to reject this paper in this version.


-------------------POST-REBUTTAL COMMENTS-------------------

I thank the authors for the response and the efforts in the updated draft. After this rebuttal, the authors well answer my comments about the comparison to SOTA methods. However, I still believe that the proposed measurement of filter importance is a limited novelty, as it is a simple revision of attention score from [Sergey Zagoruyko, ICLR 16]. Moreover, the rationale of threshold T is based on the assumption that a layer is more likely to have redundant filters to prune if it contains more remaining parameters. I think it is not a correct assumption, as the entire filters of a layer with a smaller number of parameters may be redundant to be removed safely, especially for ResNets. Thus, I still keep my original rate to reject it.

---

### Official Review · Reviewer_YdeV · 2021-11-02

**Correctness:** 3
**Technical Novelty And Significance:** 2
**Empirical Novelty And Significance:** 2
**Recommendation:** 5
**Confidence:** 3

**Main Review:**

I did not like the way that this paper is written. It is extremely waffley, and is not upfront about what is going on. It is not completely clear (to me at least) what the overall strategy for pruning will be in this work until half way through page 4 (!). The paper needs to be (completely) rewritten to match the expected format in the computer science literature.

**Strengths**
- I am not a total expert on the pruning literature, but I do find the idea of pruning based on the activation maps rather than the weights intuitive and sensible.
- The results are compelling relative to other works they compare to. NOTE: again, I will not claim to be an absolute expert on the state of the pruning literature.
- The automated approach is relatively simple, and far more interpretable than approaches such as AMC. This matters a lot in many real-world applications.
- ImageNet results are included, which is relatively rare for pruning papers.

**Weaknesses**:
- The writing of this paper is unacceptable, as I mentioned earlier.
- A few really important ablation studies are missing. Firstly, there is no direct reason to couple the pruning technique, with the method for iteratively pruning; can these not be compared separately. It is misleading to conflate the two. Secondly, there is no comment on runtime for this method, which is important if this is an automated technique.
- Another important issue is that the FLOPs reduction doesn't seem that impressive. Works like Eigendamage (Wang et al.) achieve far more impressive reductions in FLOPs, which is arguably far more important than parameters in the real world. As an aside, would you be able to compare to Eigendamage?
- The work only focuses on ResNets. I know that they're harder to prune, but I'd also like to see some experiments on VGG models, for example.

**Summary Of The Paper:**

This work proposes a technique for iterative structured pruning, without necessarily requiring too much manual human intervention. There are two parts to this paper that are important:

1. It is argued that we should prune channels based on the activation maps generated, rather than focusing on the weights of the channel.
2. They propose an iterative procedure that automatically backtracks if it has made a poor pruning decision.

**Summary Of The Review:**

I think this work has some merit, but it is not ready for publication at this time. I can be swayed at rebuttal time if there is convincing new experimental evidence.

---

### Official Review · Reviewer_EiLE · 2021-11-02

**Correctness:** 3
**Technical Novelty And Significance:** 2
**Empirical Novelty And Significance:** 2
**Recommendation:** 5
**Confidence:** 3

**Main Review:**

Strengths:
The proposed method has relatively good results on cifar10 and imagenet.  The method can automatically meet users' several requirements generally, like accuracy, latency, memory, and so on.

Weakness:
1. Each part of the technique is not that novel. It is like a combination of several existing tricks to perform comprehensive pruning, not enough analysis of the convergence or correctness. For example, how to guarantee the convergence of algorithm 3?
2. The comparisons on Imagenet are too weak. There are many SOTA pruning methods on Imagenet. The paper only lists four of them. I suggest comparing the paper with more recent SOTA pruning papers and comparing on more benchmark models besides Res50.
for example, DMCP: Differentiable Markov Channel Pruning for Neural Networks
eagleeye: fast sub-net evaluation for efficient neural network pruning.
gdp: neural network pruning via gates with differentiable polarization.
and so on.



**Summary Of The Paper:**

This paper proposed an activation-based, adaptive threshold, iterative structured pruning method, combining several existing techniques together to perform a comprehensive pruning.

**Summary Of The Review:**

Due to the concerns on the novelty and the weak comparisons on Imagenet. I temporarily think this paper is marginally below the acceptance threshold.

---

### Decision · Program_Chairs · 2022-01-20

**Decision:**

Reject

**Comment:**

The paper proposes a strategy for incrementally pruning deep learning models based on activation values. The approach can satisfy different kinds of requirements, trading off between accuracy and sparsity.

The approach seems promising and seems to have competitive performance. However, the method is described by reviewers as a combination of ideas that have been proposed in the literature, and the experimental evaluation relies too much on a dataset considered too small to be reliable in such experiments --- CIFAR10. We do not expect substantial experiments within the rebuttal period: such comparisons with relevant SOTA methods should have been present in the submission. Moreover, the strategy proposed for selecting a threshold seems to rely on some doubtful assumptions, and there are no benchmarks on actual runtime.

The writing has improved based on reviewer input, and the reviewers are satisfied with this aspect. I would still add that I would prefer some clarity in the method presentation: is there a quantity being optimized? is there a value we can monitor to ensure our reimplementation is correct? etc. In addition I would like to ask authors in the next revision to be mindful to the difference between `\citet` and `\citep` in author-year citations -- see e.g. the first two ones in 3.1.